# Moving the Needle in Gout Management: The Role of Culture, Diet, Genetics, and Personalized Patient Care Practices

**DOI:** 10.3390/nu14173590

**Published:** 2022-08-31

**Authors:** Youssef M. Roman

**Affiliations:** Department of Pharmacotherapy and Outcome Science, School of Pharmacy, Virginia Commonwealth University, Richmond, VA 23298, USA; romany2@vcu.edu

**Keywords:** acculturation, gout management, hyperuricemia, culture, race, diet, nutrition, genetics

## Abstract

Gout is a metabolic disorder, and one of the most common inflammatory arthritic conditions, caused by elevated serum urate (SU). Gout is globally rising, partly due to global dietary changes and the growing older adult population. Gout was known to affect people of high socioeconomic status. Currently, gout disproportionately affects specific population subgroups that share distinct racial and ethnic backgrounds. While genetics may predict SU levels, nongenetic factors, including diet, cultural traditions, and social determinants of health (SDOH), need to be evaluated to optimize patient treatment outcomes. This approach would allow clinicians to assess whether certain cultural norms, or some SDOH, could be contributing to their patient’s risk of developing gout or recurrent gout flares. A cultural assessment may inform the development of culturally tailored dietary recommendations for patients with gout. Causal and association studies investigating the interaction between diet, genetics, and gout, should be cautiously interpreted due to the lack of reproducibility in different racial groups. Optimal gout management could benefit from a multidisciplinary approach, involving pharmacists and nurses. While data on the effect of specific dietary recommendations on managing hyperuricemia and gout may be limited, counseling patients with gout on the role of a healthy diet to optimally control their gout flares and other comorbidities should be part of patient education. Future research investigating the role of a gene–diet interaction in the context of hyperuricemia and gout is needed. Optimal care for patients with gout needs to include a holistic assessment for gout and gout-related comorbidities. Additionally, addressing health beliefs and culture-specific lifestyle factors among patients with gout may reduce their risk of gout flare, improve adherence to urate-lowering therapy (ULT), and achieve health equity in gout management.

## 1. Introduction

Gout is a metabolic disorder, and one of the most common inflammatory arthritic conditions worldwide, caused by persistent hyperuricemia. Developing gout is multifactorial, ushering in different methodological approaches to ascertain the risk factors associated with developing hyperuricemia and gout. Despite substantial advancement in understanding the biological basis of gout, it remains one of the most poorly managed chronic conditions in healthcare. Uncontrolled gout is associated with a poor quality of life, joint damage, an increase in missed days of work, and a higher utilization of the healthcare system resources [1,2,3]. Addressing the intersection of the biological framework for developing gout and the health behaviors of patients with gout could lead to optimal gout management, through research and personalized optimal patient care practices. Therefore, the purpose of this review is to offer new insights into the development and management of gout and the implications of nongenetic factors, including diet, race, cultural beliefs, and social determinants of health (SDOH), in the health disparities of gout beyond the traditional biological signatures. Such discussion is warranted to enable researchers and clinicians to leverage recent advancements in the field and contextualize the findings of the biological basis of the various risk factors for developing hyperuricemia and gout to be personalized at the patient level. Understanding the limitations of the current biological framework of developing gout, we could collectively and holistically reduce the gout burden on marginalized and racial minorities and optimize gout management. A summary of the various predictors of gout treatment outcomes is shown in Figure 1.

## 2. Gout History and Health Disparities

Gout was historically known as the disease of kings, fueled by greater access to specific foods that only affluent people could afford. Centuries later, gout became a prevalent chronic health condition, affecting people of all social classes and various racial backgrounds [4]. In the past decades, gout and hyperuricemia prevalence shifted from becoming a socioeconomic-based disease to a pervasive chronic health condition linked to developing cardiometabolic disorders [5,6,7]. Within the realm of gout epidemiology, gout prevalence appears to disproportionately impact specific population subgroups that share distinct ancestral and cultural backgrounds. In the United States (US), gout affects 4.8% of African Americans and 2% of Hispanics [8]. In the Polynesia region, gout prevalence is the highest globally, with a rate of up to 13.9% among Pacific Islanders [9,10]. This differential gout prevalence across different racial groups is an example of health disparities compounded by intercorrelated factors, such as diet, genetics, health behavior, belief system, and SDOH [11,12]. Elucidating and addressing the causes of health disparities in gout can be complex. Potential causal factors are often intercorrelated and could exist at different individual and organizational levels, thereby necessitating a holistic approach to gout treatment and prevention. Ultimately, culturally tailored efforts focusing on clinical and non-clinical factors associated with developing gout may reduce gout-related disparities.

## 3. Global Epidemiology of Gout

The global prevalence of hyperuricemia and gout is rising, possibly due to trends in global dietary habits, the increasing global burden of gout-associated chronic diseases, the growing older adult population, obesity, and the frequent use of diuretics [13]. Specifically, gout affects more than 41 million worldwide and more than 9 million in the US [4,8]. Nonetheless, both the hyperuricemia and gout prevalence rates are higher in specific racial and ethnic groups than in European (EUR) populations [4,8,11]. With SU acting as an antioxidant conferring additional benefits to human health during major population bottlenecks, researchers postulate that certain racial groups could be inherently predisposed with hyperuricemia or gout risk alleles [11,14]. As with many other diseases, genetic predisposition to hyperuricemia and gout could modulate gout onset when coupled with critical risk factors. This hypothesis led to the suggestion that gene–diet interactions would have different outcomes based on the population studied, the consumed diet, and specific dietary enrichments among selected population subgroups. The implications of gene–diet interactions could be significantly attributed to racial and ethnic-specific variants and their respective effect sizes, leading to distinct population-level risk factors when accustomed to a purine-rich diet, such as the Western diet [15,16]. To illustrate, during the last few decades, strong evidence emerged that high-fructose corn syrup (HFCS) consumption increased substantially along with food consumption patterns toward high-calorie diets rich in sugar-sweetened beverages and processed foods containing HFCS or sucrose [17]. According to the US Department of Agriculture, the average fructose intake continuously increased from 37 g/day to 49 g/day between 1977 and 2004 [18]. The same trend was evident in non-US populations. According to the International Sugar Organization, the average sugar consumption per capita progressively increased from 56 g/day to 65 g/day between 1986 and 2007 [19]. While fructose consumption patterns could partly explain the global rise in gout prevalence, it was also reported that fructose-induced hyperuricemia is not equally distributed across different populations, due to genetic polymorphisms in the GLUT-9 encoding gene (*SLC2A9*) [15,16]. Nonetheless, there is a paucity of research investigating the effect of SU genetic polymorphisms on SU-raising or SU-lowering diets.

## 4. Gout Research Landscape

Gout is a chronic inflammatory condition caused by persistent hyperuricemia, leading to the formation and deposition of monosodium urate crystals into and around the distal joints. The development of hyperuricemia and gout is heterogenous, and, therefore, different research approaches are needed to identify and quantify the distinct risk factors in the pathogenesis of both conditions. For example, the Mendelian Randomization (MR) approach provides a pathway to ascertain causality, exploiting the natural randomization of allele causal disease. However, this approach is not without limitations, possibly due to the pleiotropic effect of the selected instrumental variables [20]. Furthermore, most MR studies have focused on the causal relationship between serum urate levels and developing cardiometabolic disorders, yet no studies have investigated the causal effect of gout on developing cardiometabolic and renal disorders. Unlike the MR approach, genome-wide association studies (GWAS) are agnostic research methods that uncover genetic signals associated with a prespecified phenotype. Despite the lucrative approach to deriving information from GWAS, these studies were predominantly conducted on EUR populations. This lack of racial diversity limits the reproducibility of their findings in non-EUR ancestry [21]. As gout continues to disproportionately affect non-EUR populations, there is a growing need to increase the representation of minorities in genetic research and cross-validation of genetic findings in multiple populations. To that end, we recognized that developing hyperuricemia and gout is a multifactorial process founded in genetics and modulated by epigenetic factors, including medications, lifestyle factors, diet, and the potential interactions between all of them. While genetic polymorphisms in *ABCG2* and *SLC2A9* remain two of the most significant signals in developing hyperuricemia and gout across different populations, evaluating nongenetic factors across selected populations through a cultural lens is an adjunct approach to further stratify hyperuricemia and gout risk and optimize gout management. This encompassing approach could be a valuable tool for gout patients with strong cultural identities and distinct racial or ethnic backgrounds. A summary of the major genes associated with regulating uric acid in humans is listed in Table 1.

## 5. Heritability of Urate Levels and Urate-Modifying Factors

Twin studies have demonstrated that serum urate (SU) levels and hyperuricemia are genetically linked with heritable estimates of 40 and 60%, respectively [22,23]. While high SU levels are strongly predictive for developing gout, not all hyperuricemia cases will result in gout, suggesting that gout is a trait influenced more by the environmental factors than the inherited factors [22]. This knowledge supports that many cases of gout could be preventable. Furthermore, specific dietary and other social and behavioral factors could significantly influence SU levels [24]. For example, social lifestyle factors such as smoking and alcohol intake could decrease and increase SU levels, respectively [25]. Health and nutritional supplements (e.g., niacin, vitamin C, cherries, and fish oil) and physical activity levels can further modulate SU concentrations and the prognostications of chronic hyperuricemia [26,27,28]. Certain medications may also affect SU levels, which warrants using or avoiding certain prescription drugs in patients with gout when compelling indications persist [29]. To that end, disproportionate disease prevalence among specific racial groups (e.g., cardiometabolic disorders in African Americans and Filipino Americans) could influence the interpretations of MR or GWAS findings by introducing tangential confounders that directly or indirectly affect SU levels. This constellation of risk factors underscores the importance of a holistic assessment of the individual’s hyperuricemia or gout risk beyond specific genetic factors and selected dietary habits to account for the disproportionate disease excess among selected population groups. Moreover, the social context of the person’s life can significantly influence the risk of exposure, degree of susceptibility, and course and outcome of diseases, such as hyperuricemia and gout. A summary of the effect of major dietary patterns and lifestyle factors on uric acid levels and gout risk is listed in Table 2.

## 6. Gout Risk and Acculturation

Dietary habits are mirrors of cultural customs and traditions. Certain population subgroups tend to follow a specific diet for personal, social, and religious norms [11,58]. Nonetheless, immigration or acculturation could significantly impact the lifestyle and dietary habits of the same population groups [58,59,60]. These changes could have significant effects on the individuals’ overall health, ranging from energy expenditure-related activities to their gut microbiome. Collectively, these changes could have a consequential impact on developing cardiometabolic risk factors, including hyperuricemia and gout. Moreover, this hypothesis is consistent with epidemiological data suggesting that gout remains a prevalent disease among developed countries, including the US. Globally, patients of Asian descent are nearly three times more likely to be diagnosed with gout than Caucasians in the ambulatory care setting in the US [61]. Similarly, Filipinos living in the US were reported to have higher gout rates and elevated means SU levels than those residing in the Philippines, suggesting significant gene–environment interactions [62,63]. Conversely, developing hyperuricemia or gout among US immigrant groups could be owing to a more permissive environment, increasing the penetrance of risk alleles by equivalent effects [58,64]. For example, the joint effect of alcohol consumption and carrying the risk allele of *ABCG2* rs2231142 G > T was associated with a greater risk for developing hyperuricemia than the risk allele alone, especially among women [65]. Therefore, ascertaining the dietary and social lifestyle habits among distinct racial and ethnic groups could shed additional light on the hypothesis of gene–diet/gene–social habits interactions and population-specific risk for developing hyperuricemia or gout [50,66]. Additionally, this ascertainment may provide more evidence on the role of preserving a cultural identity in disease prevention and the generational effect on disease onset among immigrant groups.

For example, individuals of Asian ancestry are likely to be genetically predisposed to hyperuricemia and are socially accustomed to high carbohydrate and fat and low animal protein dietary patterns [58,67]. This constellation of population characteristics could significantly increase the risk of developing gout when faced with a Westernized diet [61,68]. This is consistent with the disproportionately higher numbers of individuals of Asian ancestry seeking ambulatory care for gout visits than Caucasians [61]. While population genetics remain a key factor in gout disease, these racial health disparities further highlight the significant role of lifestyle factors in modulating the risk of developing hyperuricemia or gout [24,37,69,70]. Specifically, the compilation of a low-energy expenditure lifestyle and a high-purine diet may increase the risk of developing gout or gout-related comorbidities. Moreover, the shift from a subsistence agriculture lifestyle to a greater dependence on machine operations labor among immigrants and refugees coupled with lower-energy expenditure lifestyles and abundant purine sources while living in the US is a perfect storm for developing cardiovascular risk factors, including gout and hyperuricemia [71].

Findings from a cohort study of Hmong and Karen refugees revealed that immigration to the US is associated with changes to the gut microbiome beginning immediately upon arrival and continuing over decades. These changes included loss of gut microbiome diversity, loss of native strains, loss of fiber degradation capability, and shifts from *Prevotella* dominance to *Bacteroides* dominance, all of which may be predisposing the Hmong to metabolic diseases [72]. The study determined that only 16% of the variance could be attributed to diet and noted that other factors associated with living in the US may have adversely influenced microbiome diversity. Such results raised questions regarding intestinal changes in the uricase-producing microbiome and their effects on the risk of developing gout. A study of patients with gout (*n* = 35) compared to healthy individuals (*n* = 33) suggested that the intestinal microbiota profiles were different in both organismal and functional structures, with the intestinal microbiota of gout patients being similar to those with type-2 diabetes and metabolic syndrome [73]. Furthermore, the study showed that the reference microbial gene catalog for gout cases was consistent with disorders associated with purine metabolism and butyric acid biosynthesis, both of which are culprits for developing gout [73]. Collectively, gut microbiome composition and function, influenced by migration or found in patients with gout, are suggestive that lifestyle changes, mostly driven by acculturation, could play a significant role in the development of metabolic disorders, including gout and hyperuricemia, among immigrants who become accustomed to the Western lifestyle.

## 7. Gout Risk and Health Beliefs

Developing gout was deeply rooted in the lifestyle of excessive alcohol consumption, seafood, and red meat [74]. The framework of dietary excess for developing gout significantly contributed to the misconception of gout being a self-inflicted disease; this ingrained perception of gout rendered specific dietary restrictions to be a widely accepted gout management approach among gout patients and some healthcare professionals. This dietary framework could be contributing to poor adherence to urate-lowering therapy and fewer gout patients being prescribed urate-lowering treatments [8,75]. While there is data supporting the role of dietary-based interventions in lowering SU, dietary changes are often viewed as either a preventative or adjunct treatment approach for gout [31]. Nonetheless, the interplay between gout and developing cardiometabolic diseases may provide a window of opportunity that could be leveraged to manage gout and other existing comorbidities [76]. Collectively, this opportunity could also benefit the patient to form a healthy lifestyle extending beyond gout [76]. However, it should be recognized that access to healthy foods is not equally distributed across the population, which may hinder the effectiveness and sustainability of diet-based interventions. Therefore, patients diagnosed with gout should ideally be evaluated for other comorbidities to garner the added benefits from dietary changes. This approach will allow for patient-centered recommendations and increase the likelihood of patients adopting the provided dietary-based recommendations. Moreover, implementing novel gout management approaches embedded in real-time self-monitoring, such as home SU monitoring, may help reconstruct the dietary framework for disease management among patients with gout.

Root cause analysis of frequent hospital admissions for acute gout flares could provide additional insights to identify the barriers associated with poor disease outcomes. A qualitative semi-structured study demonstrated that treatment avoidance behaviors and recurrent gout flares could be the result of viewing gout as an insignificant disease primarily occurring in older adults [77]. Additionally, gaps in the providers’ knowledge of gout diagnosis and management, coupled with limited patient education by healthcare professionals, were also reported to impede the optimal management of gout, especially among multimorbid gout patients [77]. Among gout patients not viewing gout as a chronic disease that requires long-term treatment was identified as a barrier to adherence to ULT [78]. This perceived psychosocial burden associated with considering gout a chronic condition renders many patients reliant on hospital admissions to receive gout care.

## 8. Gout and Social Determinants of Health

Many factors could lead to ethnic and racial disparities in the prevalence and management of chronic diseases, including gout [11,58,61,79,80]. While the biomedical framework for inequality remains the operational framework to provide insights into addressing gout health disparities, little is known about the role of SDOH and gout. This role is compounded by the association between gout and multiple cardiometabolic diseases [7]. It is presumed that optimal management of gout-related comorbidities may confer added benefits for gout management itself. Therefore, optimal management of gout is believed to be within the context of the optimal chronic disease management framework. However, the management of gout remains poor, despite the availability of effective and affordable treatments. Many factors, ranging from individual to organizational causes, can lead to suboptimal management of gout. On an individual level, financial barriers, health insurance, health literacy, and access to healthy food choices are significant predictors for optimal gout management. On an organizational level, the knowledge base of gout management among healthcare professionals, adequate teaching of gout in health profession programs, patient-physician relationships, and conflicting guidelines for gout management are crucial elements for delivering optimal care for patients with gout. Assessing the role of SDOH in patients with gout may provide a window of opportunity to address treatment goals and eliminate barriers to receiving care. Furthermore, future gout studies are also needed to quantify the impact of SDOH on disease onset and treatment outcomes to inform the value of a comprehensive assessment for optimal gout management.

## 9. Multidisciplinary Approach to Gout Management

A multidisciplinary approach to chronic disease management improves treatment outcomes across many disease states. Similarly, gout management could be optimized by engaging multiple healthcare professionals in addition to physicians, including nurses, pharmacists, and dietitians. According to the 2020 American College of Rheumatology guidelines, an augmented treat-to-target protocol by nonphysician providers is conditionally recommended over the usual care [30]. Recognizing that patients with gout are more than likely to have other comorbidities, which also require close monitoring and complex treatment regimens, the pharmacist is well-positioned to optimize gout management outcomes. Indeed, pharmacist-led interventions to improve adherence to allopurinol and gout treatment had better treatment outcomes than the usual care. Specifically, a one-year pharmacist-led intervention, incorporating automated telephone technology, led to a 70% improvement in adherence rates to allopurinol and more than a two-fold increase in the number of patients achieving SU levels less than 6 mg/dL, compared with the usual care. Moreover, participants in the intervention arm were twice as likely to have their allopurinol dose escalated compared with the standard of care [81]. These results support the role of a pharmacist-staffed gout management clinic to improve gout treatment outcomes compared with the standard of care. Regardless of the clinical setting, including virtual gout clinics, pharmacist involvement in gout management significantly improves the number of individuals achieving SU levels of less than 6 mg/dL, with these patients having fewer gout flares than those receiving the standard of care [82,83]. As new patient care models continue to emerge, community-based pharmacy practice may also play a role in health equity and increasing access to healthcare among patients with chronic diseases [84].

Consistent with the benefits of a multidisciplinary approach to chronic disease management and the targeting of patient-centered outcomes, a nurse-led gout management study showed favorable results and improved gout treatment outcomes compared with the usual care. Specifically, a nurse-led intervention in the United Kingdom was associated with a high uptake of and adherence to ULT, resulting in more patients achieving SU levels of less than 6 mg/dL than patients receiving the usual care (95% vs. 30%; RR 3.18; 95% CI, 2.42–4.18) [85]. At two years, the study also showed that nurse-led gout management was associated with fewer patients experiencing two or more gout flares versus the usual care (8% vs. 24%; RR 0.33; 95% CI, 0.19–0.57) [85]. Collectively, these reports strongly support the role of nonphysicians in the gout care model to optimize patient-centered outcomes. This multidisciplinary approach can also lead to the development of a cost-effective and potentially cost-saving gout care approach.

## 10. Pharmacogenomics and Gout Management

Pharmacogenetics (PGx) is a growing field within the precision medicine era, focusing on how gene variations affect the patient’s response to treatment. Pharmacogenetics is a promising tool to optimize the selection and dosing of medications, including ULTs among patients with gout [86]. Genetic and experimental findings have demonstrated that genetic polymorphisms associated with SU pathology are also of pharmacogenetic interest [86]. Patients with gout often present with several comorbidities, warranting the use of several acute and long-term medications that increase their pill burden and the risk of adverse drug events. Implementing PGx testing can identify individuals who are more or less likely to benefit from a given treatment, thereby potentially improving medication adherence and reducing pill burden. The strongest evidence today suggests that individuals carrying the *HLA-B*58:01* allele are at a higher risk of serious and life-threatening skin reactions when taking allopurinol [30,87]. Additionally, racial disparities in the frequency of *HLA-B*58:01* warrant genetic screening in high-risk populations, specifically some Asian subgroups and African Americans [30,87]. Individuals that are G6PD-deficient can develop hemolytic anemia and methemoglobinemia with pegloticase and probenecid use [30,88]. Patients with the less active form of the drug-metabolizing CYP2C9 are at higher risk for NSAID-related upper gastrointestinal bleeding [89].

Emerging evidence of clinically significant drug–gene pairs among various gout therapies is growing. Allopurinol remains the most widely prescribed ULT; therefore, identifying sources of variability in response to allopurinol is an active area of research. Genes that were found to modulate the response to allopurinol include *AOX*, *ABCG2*, and *SLC22A12* [90,91,92,93]. Meanwhile, *UGT1A1* appears to modulate the response to febuxostat. While CYP2C9 may modulate the toxicity of benzbromarone, *SLC22A12* and ABCB1 were found to modulate the response to both benzbromarone and probenecid. The genes *CYP2D6*, *ABCB1*, gene cluster (rs6916345 G > A), and SEPHS1 were recently reported to modulate the safety and efficacy of colchicine [86,94]. Finally, *HCG22* and *IL1RN* could be linked with the response to corticosteroids and anakinra, respectively [86].

Despite the potential clinical influence that PGx could have on patient treatment outcomes, the current field is not without real-world challenges [95]. For example, the genetic testing results may not be readily interpretable; therefore, providers’ education and the involvement of different healthcare professionals are necessary. Additionally, some genetic testing may not be covered by some insurance companies, adding a financial burden to some patients; thus, advocating for patients and facilitating patients’ access to payment installment options are warranted. Integrating PGx results into the patient’s medical records and having efficient clinical decision-support tools are critical to garnering the full benefits of genomic medicine. Furthermore, the results of some PGx studies could be conflicting; accordingly, reproducible PGx studies and a greater population diversity in genetic research will be essential to delineate the distinct interpopulation genetic differences. Finally, while genetics could significantly impact the response to a given treatment, other factors could also modulate the overall response to therapy. A summary of the potential predictors in response to treatment is shown in Figure 2.

## 11. Roadmap to Optimize Gout Management

Indeed, genetics may significantly predict SU levels; nonetheless, nongenetic factors, including diet, health beliefs, and SDOH, need to be evaluated through a cultural lens and by culturally competent healthcare providers [96]. This approach would allow clinicians to assess whether certain cultural norms or specific health barriers could be contributing to their patient’s risk of developing gout or experiencing recurrent gout flares [58]. For example, a one-year follow-up study of Māori and Pacific Islander patients with gout identified that Māori and Pacific Islander ethnicity was independently associated with a higher gout flare frequency and reduced physical functioning at the baseline and after one year [97]. Therefore, being cognizant of the patient’s ancestry and cultural background may help provide targeted care and address the population-specific risks. This cultural assessment approach may also inform the development of culturally tailored dietary recommendations for patients with gout. While the field of nutrigenomics is an exciting path to accomplishing precision health, results from the studies investigating the role of diet and genetics should be cautiously interpreted, possibly due to the lack of reproducibility in different racial groups, especially underserved and minority population subgroups. To this end, results from GWAS warrants reproducibility and validation across distinct population subgroups to become clinically actionable. While data on the effect of specific dietary approaches to manage hyperuricemia and gout may be forthcoming, counseling patients with gout on the role of a healthy diet to optimally control gout flares and other comorbidities should be part of patient education. Ultimately, the studies assessing optimal care delivery for gout patients need to examine the potential benefits of conducting a holistic assessment for gout and gout-related comorbidities. Furthermore, addressing culture-specific lifestyles and patient-specific SDOH may significantly mitigate their risk for gout flare and improve their adherence to ULT. Educating healthcare community providers on the chronic disease approach to gout management should be part of a multipronged approach, including patient education by a healthcare team member, a shared decision-making process, and the clinical infrastructure to achieve the desired treatment outcomes. Emulating the 2020 American College of Rheumatology guidelines, a proposed roadmap for optimal gout management is summarized in Figure 3.

## 12. Conclusions

The prevalence of gout is rising globally, and disproportionality affects distinct racial and ethnic groups. Although the development of gout is multifactorial, with a strong link to genetics, other factors could be contributing to the disproportionate disease prevalence. While significant advancements have been made in characterizing the genetic risk factors associated with developing hyperuricemia and gout, the disease remains one of the most suboptimally managed chronic conditions in our healthcare system, urging the need for innovative solutions to address gout management gaps. Addressing gout management deficiencies requires an encompassing approach that emphasizes increasing the representation of racial and ethnic minorities in genetic research, leveraging a multidisciplinary approach for disease management, developing gout management core competencies for providers, and conducting a holistic and culturally tailored patient-centered management approach. Although dietary interventions alone may not replace the need for urate-lowering therapy, it remains a window of opportunity to comprehensively address the cardiometabolic disorders among patients with gout.

## Figures and Tables

**Figure 1 nutrients-14-03590-f001:**
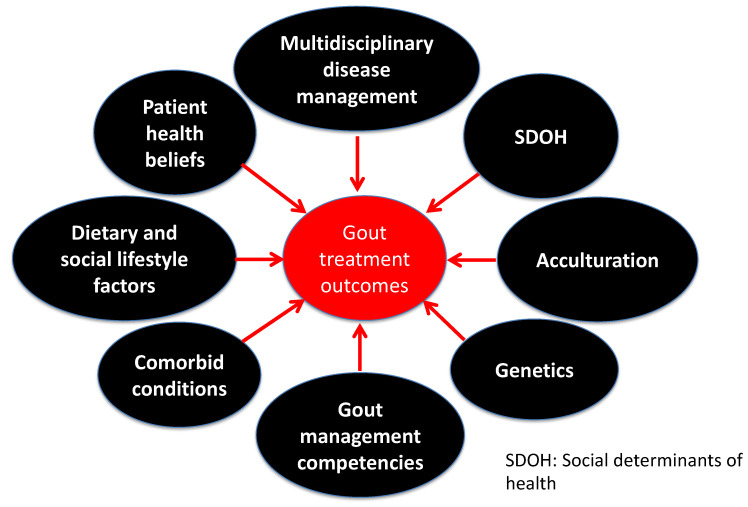
Predictors of gout treatment outcomes.

**Figure 2 nutrients-14-03590-f002:**
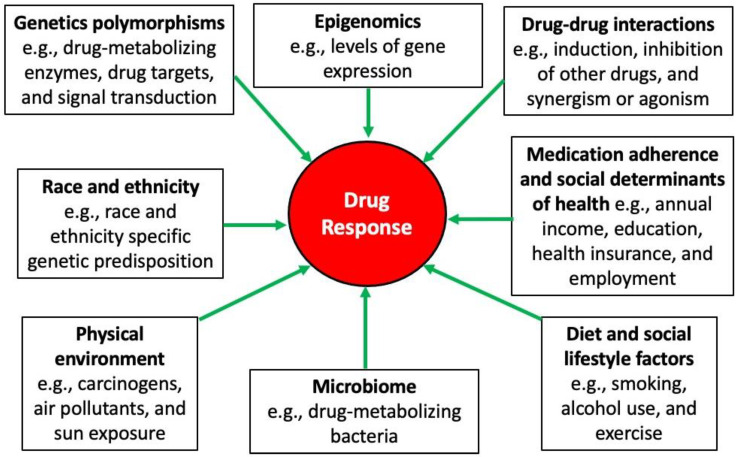
Sources of variability in drug response [58].

**Figure 3 nutrients-14-03590-f003:**
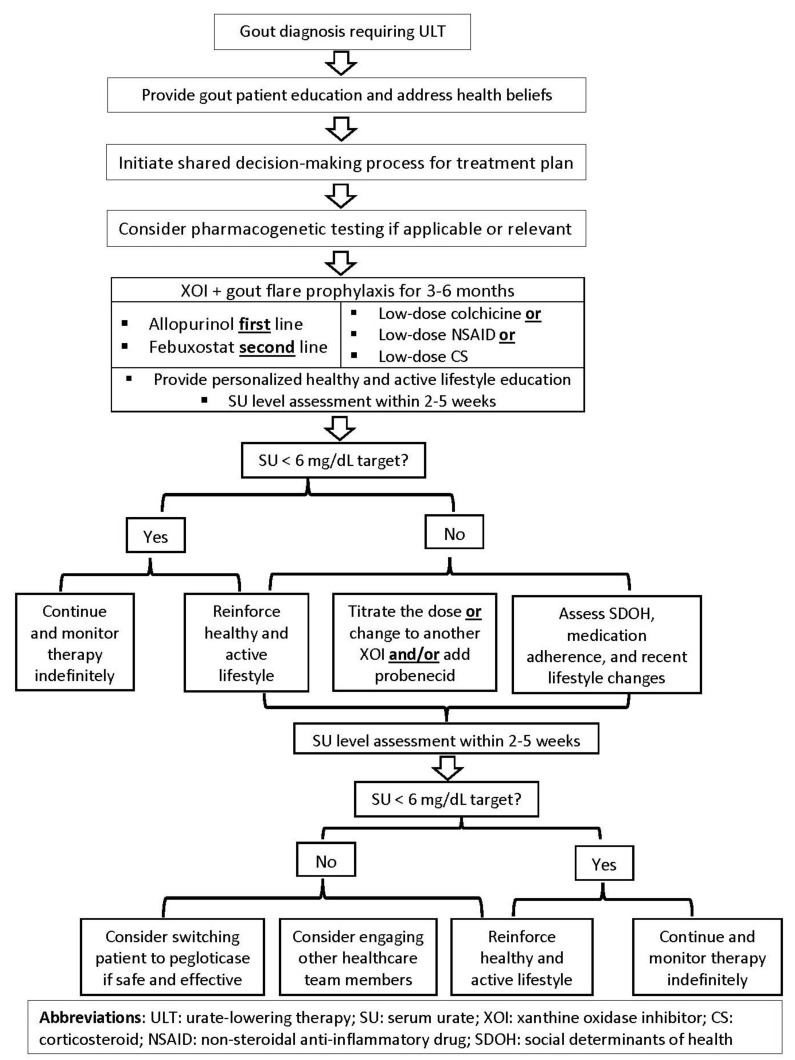
Proposed roadmap for optimal gout treatment in adults.

**Table 1 nutrients-14-03590-t001:** Summary of major urate regulation genes.

Gene	Protein	Possible Functions
*ABCG2*	ATP binding cassette subfamily G member 2: ABCG2	Regulating renal and gut excretion of urate. Gene polymorphisms are strongly linked to urate underexcretion and the risk of early-onset gout in men. Genetic polymorphisms may also influence the therapeutic response to allopurinol and other statin medications.
*GCKR*	Glucokinase regulator	Regulatory protein that inhibits glucokinase in the liver and pancreatic islet cells by forming an inactive complex with the enzyme. Gene polymorphisms are associated with fasting glucose, maturity-onset type-2 diabetes, hyperuricemia, and gout.
*LRRC16A*	Capping protein regulator and myosin 1 linker 1: CARMIL1	Cytoskeleton-associated protein. Gene polymorphisms are associated with urate concentrations and gout subtypes.
*PDZK1*	PDZK domain-containing scaffolding protein	Mediates the localization of cell surface proteins and plays a critical role in cholesterol metabolism. Gene polymorphisms are linked to dyslipidemia, hyperuricemia, and gout.
*SLC2A9*	Solute carrier family 2 member 9: GLUT9	Regulating renal uric acid reabsorption. Gene polymorphisms are linked to the risk of gout in women.
*SLC16A9*	Solute carrier family 16 member 9: MCT9	Regulating monocarboxylic acid transporter. Gene polymorphisms are linked to uric acid concentrations.
*SLC17A1*	Solute carrier family 17 member 1: NPT1	Sodium phosphate cotransporter. Gene polymorphisms are linked with hyperuricemia and gout.
*SLC22A11*	Solute carrier family 22 member 11: OAT4	Urate reabsorption transporter. A target for some uricosuric drugs. Gene polymorphisms are associated with hyperuricemia.
*SLC22A12*	Solute carrier family 22 member 12: URAT1	Uric acid reabsorption transporter. A major target for uricosuric drugs. Gene polymorphisms are associated with hyperuricemia and gout. Loss of function in the gene can also lead to hypouricemia.

**Table 2 nutrients-14-03590-t002:** Effect of dietary patterns and lifestyle factors on serum urate and gout risk management.

Diet/Food/Lifestyle Factor	Serum Urate Level	Incident Gout	Gout Flare Risk	ACR 2020 Recommendations [30]	References
DASH diet	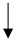	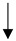	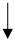	No recommendation	[31,32,33]
Mediterranean diet	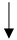	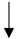	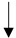	No recommendation	[34]
Ketogenic diet	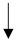	No data	No data	No recommendation	[35]
Low-fat dairyproducts	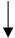	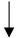	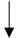	No recommendation	[36,37]
Cherries	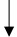	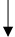	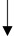	No recommendation	[38,39]
Coffee	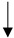 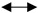	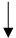	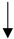	No recommendation	[40,41,42,43]
Tea	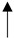 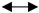	No data	No data	No recommendation	[42,43,44]
High-fructose corn syrup (HFCS)	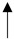	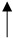	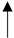	Conditionally recommends limiting the intake of HFCS	[15,19]
Weight loss	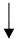	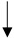	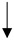	Conditionally recommends a weight loss program	[45,46]
Physical exercise	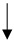	No data	No data	No recommendation	[26,45]
Smoking	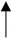 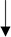	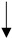	No data	No recommendation	[47,48,49]
Alcohol	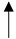	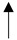	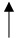	Conditionally recommends limiting alcohol intake	[50,51,52]
Vitamin B complex (B6-B12-Folic acid)	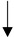	No data	No data	No recommendation	[53]
Vitamin C	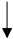 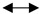	No data	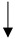	Conditionally recommends against use	[27,54,55]
Fish Oil/Omega-3-fatty acids	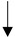	No data	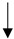	No recommendation	[28,56,57]

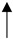
, increased; 
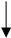
, decreased; 
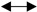
 clinically insignificant or no effect.

## Data Availability

Not applicable.

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
