# Peer review of "Moving the Needle in Gout Management: The Role of Culture, Diet, Genetics, and Personalized Patient Care Practices"

_nutrients, 2022, doi:10.3390/nu14173590_

Round 1

Reviewer 1 Report

The author collected a lot of related gout-related articles and pointed out the possible effects in many aspects, which is worthy of encouragement. The author pointed out that this narrative-oriented content is a bit too short, and many do not understand the article's content. But the author deserves credit for putting together a table of numerous articles to get to the point. The resolution of Figure 3 is too low, the words are very fuzzy, and many overlapping parts within the box need to be remapped to meet the journal requirements.

Author Response

Thanks for the kind and positive feedback. I revised figure 3 to address the concerns raised by reviewer 1. I also tried to revise certain parts of the content to increase the overall understanding of the manuscript. 

Reviewer 2 Report

The multifaceted nature of the publication, taking into account genetic, environmental and nutritional factors as well as global epidemiological risk factors. The figures are a valuable supplement to the issues presented in the publication. The practical aspect of the study is highlighted by the “roadmap for optimal gout treatment in adults”.

Remarks

I propose to present a part of the publication "Gout History and Health Disparities" immediately after the part "Introduction"

Author Response

Thanks for the kind comments and feedback. I also moved the sections of Gout History and Health Disparities following the introduction section. 

Round 2

Reviewer 1 Report

The author has made revisions in accordance with the previous requirements and redrawn the figure. This review paper can be accepted.